# Essential Oils of *Alpinia nantoensis* Retard Forskolin-Induced Melanogenesis via ERK1/2-Mediated Proteasomal Degradation of MITF

**DOI:** 10.3390/plants9121672

**Published:** 2020-11-28

**Authors:** K. J. Senthil Kumar, M. Gokila Vani, Pei-Chen Wu, Hui-Ju Lee, Yen-Hsueh Tseng, Sheng-Yang Wang

**Affiliations:** 1Department of Forestry, National Chung Hsing University, Taichung 402, Taiwan; zenkumar@dragon.nchu.edu.tw (K.J.S.K.); mgvani2009@gmail.com (M.G.V.); cbevh99@yahoo.com.tw (P.-C.W.); ab840909@yahoo.com.tw (H.-J.L.); tsengyh2014@gmail.com (Y.-H.T.); 2Center for General Education, National Chung Hsing University, Taichung 402, Taiwan; 3Agricultural Biotechnology Research Institute, Academia Sinica, Taipei 115, Taiwan

**Keywords:** *Alpinia nantoensis*, Zingiberaceae, essential oil, anti-melanogenesis, forskolin, MITF

## Abstract

The anti-melanogenic activity of essential oils of *Alpinia nantoensis* and their bioactive ingredients were investigated in vitro. Treatment with leaf (LEO) and rhizome (REO) essential oils of *A. nantoensis,* significantly reduced forskolin-induced melanin production followed by down-regulation of tyrosinase (TYR) and tyrosinase related protein-1 (TRP-1) expression at both transcriptional and translational levels. Further studies revealed that down-regulation TYR and TRP-1 were caused by LEO/REO-mediated suppression of Microphthalmia-associated transcription factor (MITF), as evidenced by reduced nuclear translocation of MITF. Also, we found that LEO/REO induce the sustained activation of ERK1/2, which facilitate subsequent proteasomal degradation of MITF, as confirmed by that LEO/REO failed to inhibits MITF activity in ERK1/2 inhibitor treated cells. In addition, a significant increase of ubiquitinated MITF was observed after treatment with LEO and REO. Furthermore, the chemical composition of LEO and REO were characterized by gas chromatography-mass spectrometry (GC-MS) resulted that camphor, camphene, α-pinene, β-pinene, isoborneol and _D_-limonene were the major compounds in both LEO and REO. Further studies revealed that α-pinene and _D_-limonene were the active components responsible for the anti-melanogenic properties of LEO and REO. Based on the results, this study provided a strong evidence that LEO and REO could be promising natural sources for the development of novel skin-whitening agents for the cosmetic purposes.

## 1. Introduction

Melanin is a mixture of pigmented biopolymers synthesized by specialized cells determinant of skin, hair and eye color. Although pigmentation play a crucial photo-protective role against the carcinogenetic effects of ultra violet (UV) irradiation through direct UV absorption, excessive production of melanin causes hyperpigmentary skin disorders, including melisma, Riehl melanosis, erythromelanosis, follicularis faciei, poikiloderma of civatte, porphyria cutanea and melanoma [1]. Melanin biosynthesis in melanocytes is a complex process involved many factors, including UV irradiation, stem cell factors (SCF), α-melanocyte stimulating hormone (α-MSH), cAMP and transcription factors [2]. For example, α-MSH binds to melanocortin receptor-1 (MC1R), a member of G-protein receptors in melanocytes induces the activation of adenylyl cyclase enzyme, followed by increased cyclic adenosine monophosphate (cAMP) production. cAMP leads to transcriptional activation of cAMP response element-binding protein (CREB), which in turn stimulate microphthalmia transcription factor (MITF) promoter activity [3]. MITF, a basic leucine zipper transcription factor directly binds to the promoter regions of melanin production genes and positively regulates their genes, including tyrosinase (*TYR*), tyrosinase related protein-1 (*TRP-1*) and dopachrome tautomerase (*DCT*) [3]. Among these enzymes, tyrosinase is the rate-limiting enzyme and catalyzes the hydroxylation of tyrosinase to _L_-3,4-dihydroxyphenylalanine (_L_-DOPA) and the oxidation of _L_-DOPA to dopaquinone. DCT and TRP-1 were involved in conversion of DOPA-chrome into 5,6-dihydroxyindole-2-carboxilic acid (DHICA) and DHICA into eumelanin, respectively. There are several factors negatively regulate MITF activity in melanocytes. It was reported that the inhibition of the ERK signaling pathway induces melanoma cell proliferation and MITF activity in vitro suggesting that the ERK signaling pathway negatively regulate MITF-mediated melanogenesis [4]. Further studies have revealed that ERK phosphorylates MITF at serine 73 and that the phosphorylation of MITF at serine 73 is responsible for MITF ubiquitination and degradation [5]. Likewise, inhibition of phosphoinositide 3-kinase (PI3K) increases MITF transcriptional activity, leading to increase tyrosinase and TRP-1 expression and melanogenesis [6]. However, UV-irradiation-induced p38 MAPK activation has been shown to be involved in melanogenesis through the increase of MITF transcriptional activity and subsequent increase of tyrosinase [7].

Zingiberaceae is one of the largest dietary/spice families in the plant kingdom, which comprises 53 genera along with 1600 know species. Among them genus *Alpinia* is the largest genus in Zingiberaceae with about 230 species, which are widely distributed in tropical and sub-tropical regions of Asia, Australia and the Pacific Islands [8]. *Alpinia*, also popularly known as “shell ginger” plants were economically important and used as food, food adjunct, spices, folk medicine and cultivated ornamentals [9]. *A. nantoensis* F.Y. Lu & Y.W. Kuo is a newly identified species, which is native to the mountain regions of Taiwan. Previously, *A. nantoensis* was recognized as *A. pricei* due to its identical leaf size and flowers to those of *A. pricei*. However, its flowers has a bracteole and an oblong labellum, which differs from its congeneric native species of *A. pricei*, which has no bracteole and has a rhombic labellum (Kuo et al., 2008). Traditionally, the leaves of *A. pricei* were used to wrap zongzi (a glutinous rice dumplings) and the aromatic rhizome was used to treat abdominal discomfort and to increase stomach secretion and peristalsis (Tsao et al., 2019). While, leaves and rhizomes of *A. zerumbet* traditionally used for skin care and insect repellent (Chompoo et al., 2012). Since, there undistinguishable similarity between *A. pricei* and *A. nantoensis*, the leaves *A. nantoensis* was used to prepare zongzi and the rhizomes were used for traditional Chinese medicine preparation [10,11]. We recently reported that leaf, stem and rhizome extracts of *A. nantoensis* exhibited strong anti-metastatic properties in human breast cancer cells [11]. Another study also shown that *trans*-3-methoxy-5-hydroxystilbene isolated from the rhizome of *A. nantoensis* inhibited lung cancer cell metastasis in vitro [10]. However, other biological activities of this newly identified species was poorly investigated. Therefore, in the present study, we aimed to investigate the anti-melanogenic properties of essential oils obtained from leaf and rhizomes of *A. nantoensis*.

## 2. Results

### 2.1. Effect of LEO and REO on Melanin Production and Tyrosinase Activity

Prior to examine the anti-melanogenic properties of LEO and REO, cytotoxicity on various dermal cell lines, including murine melanoma (B16-F10), human skin fibroblast (CCD966SK), human skin keratinocytes (HaCaT) and human epidermal melanocytes-adult (HEM-a) were examined. These cell lines were exposed to increasing concentrations of LEO and REO (6.25, 12.5, 25, 50 and 100 μg/mL) for 48 h and the cell viability was determined by MTT assay. Both LEO and REO were did not show any reduction in cell viability at the concentrations used, indicating that LEO and REO were not cytotoxic to the dermal cells (Appendix A). As shown in Figure 1A, cellular melanin content was significantly up-regulated in cells treated with 20 μM FSK. Whereas, co-treatment with 100 μg/mL of either LEO or REO significantly inhibited the FSK-mediated increase of melanin content in B16-F10 cells. Notably, REO exhibited strong inhibitory effect than the LEO treated cells. Also, the results indicate that 100 μg/mL REO has stronger inhibitory effects than arbutin (100 μM) or kojic acid (100 μM), are known anti-melanogenic agents. In addition, treatment with LEO and REO resulted in a significant and dose-dependent decrease of melanin production (Appendix A). These results suggest that LEO and REO may have anti-melanogenic properties. Tyrosinase is a rate-limiting enzyme play a functional role in melanin biosynthesis. Therefore, next we examined the effect of LEO and REO on mushroom tyrosinase activity in a cell-free system. Result showed that either LEO or REO failed to inhibit mushroom tyrosinase activity, while ascorbic acid (100 μM) and kojic acid (100 μM) showed potent inhibition (Figure 1B). This data suggested that LEO and REO does not inhibit tyrosinase activity directly.

Several anti-melanogenic agents were as not direct tyrosinase inhibitors, instead they down-regulate the expression levels of tyrosinase and its related proteins, including TRP-1 and DCT by modulating cellular signaling pathways [12]. Therefore, next we determined the effect of LEO and REO on cellular tyrosinase activity in FSK-stimulated cells. Measurement of cellular tyrosinase showed that cells exposed to FSK remarkably increased cellular tyrosinase activity, whereas co-treatment with 100 μg/mL of either LEO or REO potently inhibited such activity (Figure 1C). In addition, this effect was observed in dose-dependent manner (Appendix A). To further elucidate the mechanism of inhibition, we determined the mRNA expression levels of melanogenesis regulatory genes, including *tyrosinase*, *TRP-1* and *DCT* in FSK-stimulated cells. Q-PCR analysis resulted that compared with FSK-treated cells, the mRNA expression levels of *tyrosinase* (Figure 1D), *TRP-1* (Figure 1E) and *DCT* (Figure 1F) were significantly down-regulated following co-treatment with either LEO or REO for 6 h. In addition, both LEO and REO substantially decreased the FSK-mediated increase of tyrosinase (Figure 1G), TRP-1 (Figure 1H) and DCT (Figure 1I) protein levels at 24 h.

### 2.2. LEO/REO Inhibit Melanogenesis via Suppressing MITF Transcriptional Activity

Tyrosinase and its related genes (*TRP-1* and *DCT*) are known to be transcribed by MITF, a basic helix-loop-helix leucine zipper transcription factor. Therefore, initially we examined whether LEO/REO modulate protein and mRNA levels of MITF in FRK-stimulated cells. We found that treatment with FRK significantly increased up-regulated *MITF* mRNA expression in B16-F10 cells, whereas co-treatment with LEO and REO significantly down-regulated such expression (Figure 2A). In addition, treatment with LEO and REO significantly reduced the MITF protein levels in FRK-treated cells (Figure 2B). Next, we examined we examined the nuclear export of MITF using immunofluorescence. As shown in Figure 2C, compared with control cells, an increased nuclear export of MITF was observed in FSK-stimulated cells as evidenced by accumulation of MITF proteins in the nucleus. Interestingly, co-treatment with either LEO or REO significantly blocked FSK-induced nuclear export of MITF. To further delineate the role of LEO/REO-mediated inhibition of MITF and melanogenesis, cells were transiently transfected with MITF siRNA for 6 h and then the cells were incubated with FSK in the presence or absence of LEO/REO for 48 h. Transient transfection of MITF specific siRNA significantly decreased melanin production in FSK-treated cells. A similar inhibition was also observed in LEO or REO treated cells, which was further declined by a combination with either LEO or REO (Figure 2D). These results suggest that LEO and REO inhibit melanogenesis by suppressing MITF signaling pathway.

### 2.3. LEO/REO-Mediated Down-Regulation of MITF Is Not Associated with CREB Activity

It is well established that treatment with FSK, a direct cAMP activator, triggers cAMP production and activate the CREB transcription factor via PKA. The activated CREB export to the nucleus and induces MITF promoter activation for melanin production [2]. Therefore, we examined the effect of LEO/REO on FSK-mediated CREB activity in B16-F10 cells. As shown in Figure 2E, cells exposed to FSK significantly increased CREB phosphorylation, whereas co-treatment with either LEO or REO did not inhibit the phosphorylation of CREB. In a similar way, compared with control cells, FSK treatment remarkable increased nuclear translocation of CREB as evidenced by increased accumulation of CREB in the nucleus, either LEO or REO failed to inhibit the FSK-mediated nuclear translocation of CREB (Figure 2F). Notably, co-treatment with gallic acid, a known de-pigmenting agent significantly decreased the nuclear translocation of CREB in B16-F10 cells. These data suggesting that LEO/REO-mediated inhibition of MITF activity was not associated with CREB.

### 2.4. LEO and REO Suppress MITF Activity through ERK Activation

ERK1/2 and PI3K/AKT were demonstrated as key regulators of melanogenesis through their regulation of MITF activation. Activation of ERK1/2 and PI3K/AKT leads to MITF degradation via ubiquitination, which plays a pivotal role in suppressing melanogenesis [13,14]. Therefore, we sought to examine whether LEO and REO regulate MITF activity via ERK1/2 or PI3K/AKT pathways. Western blot analysis showed that treatment with LEO and REO significantly as well as time-dependently increased ERK1/2 and AKT phosphorylation (Figure 3A). Indeed, after 15 min of treatment with LEO/REO peaked ERK1/2 phosphorylation, which was gradually decreased. Likewise, either treatment with LEO or REO immediately increased AKT phosphorylation and the increase was observed in a time-dependent manner and a profound AKT phosphorylation was noted at 6 h after treatment. However, the total levels of ERK1/2 and AKT were unaffected by either LEO or REO. According to this data, we hypothesis that LEO and REO-mediated ERK1/2 and AKT phosphorylation could possibly induce proteasome degradation of MITF. As we expected, treatment with either LEO or REO failed to inhibit FRK-induced MITF expression in ERK1/2 inhibitor (PD98059) pre-treated cells, whereas LEO and REO significantly inhibited MITF expression in PI3K/AKT inhibitor (LY294002) pre-treated cells (Figure 3B). To further delineate, the effects of LEO and REO on FSK-induced melanin production under pharmacological inhibition of ERK1/2, JNK/SAPK, p38 MAPK and PI3K/AKT were determined. A remarkable increase of melanin production (higher than FSK treated cells) was observed in ERK1/2 inhibitor-treated cells, while the melanin production was unaffected by either treatment with JNK/SAPK or p38 MAPK or PI3K/AKT in FSK-treated cells. Interestingly, co-treatment with LEO and REO failed to inhibit FSK-induced melanin production in ERK1/2 inhibitor-treated cells, whereas significant reduction in melanin production was observed in JNK/SAPK, p38 MAPK and PI3K/AKT inhibitor treated cells (Figure 3C). It was well demonstrated that agents induce ubiquitination and proteasomal degradation of MITF via phosphorylation of MITF at serine 73 residue through ERK activation [15]. To corroborate that LEO/REO-induced proteasomal degradation of MITF was caused by ubiquitination, the MITF ubiquitination was determined by immunoprecipitation. As shown in Figure 3D,E, an accelerated MITF ubiquitination was observed in LEO and REO or MG132-treated cells when compared with un-treated control cells, while treatment with ERK1/2 inhibitor decreased MITF ubiquitination to below the control level even though in the presence of LEO/REO or MG132. Taken together, these results demonstrate that LEO and REO treatment leads to ubiquitination and proteasomal degradation of MITF through ERK1/2 activation, which eventually suppress melanin production in melanoma cells.

### 2.5. Chemical Compositions of Leaf and Rhizome Essential Oils of A. nantoensis

The yields of leaf and rhizome essential oils obtained by hydro-distillation were 0.11% (*w*/*w*) and 0.20% (*w*/*w*), respectively. The major chemical constituents of leaf and rhizome essential oils and their relative amounts were determined by GC–MS analysis. The GC–MS profiles and the major compounds of LEO and REO are shown in Appendix A. The relative contents (%) in LEO and REO are shown in Table 1. A total of 15 compounds were identified in REO, accounting for 70.82% of the whole oil. The major components in LEO were camphor (21.31%), camphene (11.41%), β-pinene (8.28%), *p*-cymene (5.25%), α-pinene (4.49%) and _D_-limonene (2.5%), which made up around 53.24% of the content of the LEO. REO contains 14 identified compounds, accounting for 77.91% of the whole oil. The major compounds in REO were found to be camphor (29.58%), camphene (10.45%), β-pinene (9.13%), α-pinene (4.23%), *p*-cymene (3.22%) and _D_-limonene (2.43%), which represents 59.04% of the whole oil.

### 2.6. Effects of Major Constituents in LEO and REO on Melanin Production

The cytotoxic effect of camphor, camphene, *p*-cymene, α-pinene, β-pinene and _D_-limonene were determined by MTT assay. Treatment with either camphor or camphene or *p*-cymene or α-pinene or β-pinene did not display cytotoxicity in B16-F10 cells up to a concentration of 100 μM for 48 h, whereas _D_-limonene displayed cytotoxicity over a concentration of 50 μM (Appendix AE). To further evaluate the bioactivity compounds LEO and REO on the alteration of melanogenesis, we examined the melanin production and cellular tyrosinase activity in FRK-stimulated B16-F10 cells. Interestingly, the FRK-mediated increase of melanin content was significantly inhibited by α-pinene (100 μM) and _D_-limonene (50 μM), as these compounds reduced the melanin content from 5.45 μg/mL (FRK-treatment) to 3.73 μg/mL and 2.99 μg/mL, respectively (Figure 4A), whereas, treatment with indicated concentration of camphor, camphene and *p*-cymene were not altered the FRK-induced melanin content in B16-F10 cells. In addition, treatment with β-pinene decreased melanin production, however the decrease was not statistically significant. As shown in Figure 4B, the results of cellular tyrosinase activities were also similar to that of melanin content. Although, camphor had some inhibitory effect on cellular tyrosinase activity, however the inhibitory effect was not as significant compared with those of α-pinene and limonene treated cells. Indeed, α-pinene and _D_-limonene-mediated melanin (Figure 4C,D) and tyrosinase inhibition was also observed in a dose-dependent manner (Figure 4E,F).

## 3. Discussion

Several synthetic agents are use in the cosmetic industry as skin whitening agents. Most of these agents are direct tyrosinase inhibitors but the development of natural agents is becoming more important due to the disadvantages of synthetics such as high cytotoxicity, insufficient penetrating power and low activity. For example, kojic acid, a synthetic tyrosinase inhibitor is widely used as a skin-whitening agent in cosmetics, later it was reported to cause several serious side effects, including erythema and contact dermatitis [16]. Therefore, the identification of safe, bio-combatable as well as natural derived skin-whitening agents represents the hour of need.

Essential oils has been widely used in the preparation of pharmaceutical and cosmetic products as they offer broad-spectrum of health benefits as well as preservatives. Their biological activities are ranging from anti-bacterial, anti-fungal, anti-viral, anti-septic, analgesic, anti-inflammatory and dermato-protection [17]. In addition to their putative bioactivities, their unique pleasant aroma enables to use in cosmetic products. Currently, essential oils are one of the subjects of intensive scientific research and also received much attention from cosmetic and pharmaceutical industries. Particularly, natural product like essential oils were reported to have direct tyrosinase inhibition, which enable them to use as skin-lightening agents [18]. *Alpinia* plants are generally aromatic due to their rich content of essential oils. Previous studies have reported that aqueous extract of rhizome of *A. officinarum* and fruit extracts of *A. galangal* exhibited strong anti-melanogenic activity in cultured melanoma cells [19,20]. Therefore, we hypothesis that the essential oils of *A. nantoensis* may possess anti-melanogenic properties.

Melanin producing murine melanoma B16-F10 cell line is widely used to investigate anti-melanogenic properties of synthetic or natural agents. There are several agents induce melanogenesis in melanoma cells such as α-MSH, β-MSH, 3-isobutyl-1-methylxanthine (IBMX), forskolin (FRK), α-lipotropin (β-LPH), β-endorphin (β-END) and so forth. Among them, α-MSH, β-MSH, IBMX and forskolin (FRK) were recognized as cAMP activators, which trigger melanogenesis [21]. In the present study, we subjected FRK to induce melanin production in B16-F10 cells and determined the inhibitory effect of LEO and REO. Our results showed that co-treatment with LEO and REO significantly inhibited FRK-induced melanin production. Melanin biosynthesis is critically regulated by melanogenic enzymes, including tyrosinase, TRP-1 and DCT [21]. Therefore, we sought to determine whether LEO and REO could modulate tyrosinase enzyme activity. Since, the mushroom tyrosinase inhibitory assay is a widely used tool for determining the skin whitening effect of candidate agents in cell-free system, because tyrosinase is the limiting enzyme in melanin formation in skin. Utilizing this assay, we determined the tyrosinase inhibitory effect of LEO and REO where _L_-DOPA was used as substrate. We found that either LEO or REO failed to inhibit mushroom tyrosinase activity in cell-free system. Indeed, treatment with LEO and REO exhibited strong cellular tyrosinase activity in FRK-stimulated cells, which is consistent with other’s observation that several herbal extracts inhibits cellular tyrosinase activity without affecting mushroom tyrosinase activity [22]. Our results show that LEO and REO down-regulated the expression levels of TYR and TRP-1 possibility by suppressing MITF transcriptional activity. In contrast, a reduced expression of DCT was observed in FRK-treated cells, while LEO and REO significantly provoked FRK-mediated decrease of DCT. It has been explained by transient transfection assays that MITF isoforms regulate the transcription of DCT gene in a different manner as those of TYR and TRP-1 [23]. Thus, we speculate that LEO/REO may regulate the expression of *DCT* gene in a different manner then that of *TYR* and *TRP-1* genes, suggesting that the anti-melanogenic effect driven by LEO and REO are mediated via TYR and TRP-1, since TYR is necessary for melanogenesis as the critical rate-limiting enzyme.

During the melanogenesis, MITF is regulated at both transcriptionally and post-translationally. At the transcription level, MITF is regulated by CREB, a cAMP-dependent transcription factor transcribed MITF upon stimulation [2]. We found that LEO and REO treatment significantly down-regulated MITF protein and mRNA expression in FRK-stimulated cells, which allowed us to determine the involvement of CREB in MITF regulation. Treatment with either LEO or REO does not modulate the FRK-induced CREB phosphorylation as well as nuclear translocation. Therefore, we hypothesis that LEO/REO regulate MITF at post-translationally. In such conditions, MITF may undergo various post-translational modifications, including phosphorylation, sumoylation and ubiquitination [2]. A previous study reported that most melanoma cells contain hyper-activated MAPKs, which triggers MITF phosphorylation at serine 73, leads ubiquitin-mediated proteasomal degradation [5]. To validate our hypothesis, cells were pre-treated with pharmacological inhibitors of p38 MAPK, ERK1/2, JNK/SAPK and AKT and then exposed to FRK in the presence or absence of LEO and REO and the melanin production was determined. Interestingly, treatment with LEO and REO failed to inhibit FRK-induced melanin production in ERK1/2 inhibitor treated cells, while significant reduction in melanin production were observed in p38 MAPK, JNK/SAPK and AKT inhibitors treated cells. This data is consistent with a significant increase of ERK1/2 phosphorylation by LEO and REO. In this study, we also noted a significant and time-dependent increase of AKT phosphorylation by LEO and REO. It has been well characterized that PI3K/AKT pathway plays a key role in cellular metabolism, proliferation, survival and chemo-resistance [24]. Therefore, we speculate that the LEO and REO-mediated AKT phosphorylation may involve in cellular metabolism or survival and it is very clear that LEO/REO-mediated AKT activation does not have any functional role in MITF regulation. Furthermore, we found that treatment with LEO/REO, ERK1/2 inhibitor (PD98059) and proteasome inhibitor (MG132) were accelerated MITF degradation via ubiquitin-proteasome system. A similar anti-melanogenic mechanism was also reported by other researchers [25].

In the present study, we also investigated the chemical composition and their melanin inhibitory effect in vitro. GC-MS analyses resulted with 18 identified compound in both LEO and REO, most of these constituents are monoterpenoids and sesquiterpenoids. LEO and REO exhibited the most identical compositions, among the 18 identified compounds 12 compounds are present in both essential oils. That may be a reason for the similar effects observed in both essential oils. Further studies revealed that the major compounds, that is, camphor, camphene and *p*-cymene were not effect in the context of melanin inhibition, which is in agreements with a previous study that camphor isolated from the essential oil of *Achillea millefolium* does not showed significant melanin inhibition up to a concentration of 400 μM [26]. Indeed, the minor compounds in LEO and REO, α-pinene and _D_-limonene, exhibited strong melanin inhibition. A previous study support our notion that α-pinene is a potential anti-melanogenic agent [27]. However, to the best of our knowledge, there is no specific record exist on the anti-melanogenic action of _D_-limonene, which resulted in as a potent melanin inhibitor then that of α-pinene.

## 4. Materials and Methods

### 4.1. A. nantoensis and Essential Oil Preparation

*A. nantonensis* was collected in March 2019 from Nantou County, Taiwan and was identified by Prof Yen- Hsueh Tseng (Department of Forestry, National Chung Hsing University, Taichung, Taiwan). The voucher specimen (TCF Tseng4590) was deposited in the herbarium of the same university. Air-dried rhizome and leaves of *A. nantonensis* were subjected to hydrodistillation in a Clevenger-type apparatus for 6 h, followed by determination of oil contents. Leaf (LEO) or rhizome (REO) essential oils were stored in airtight sample vial prior to analysis by gas chromatography–mass spectrometry (GC–MS) and bioactivity evaluation.

### 4.2. Chemicals and Reagents

Roswell Park Memorial Institute *(*RPMI*)*
*1640 medium,* fetal bovine serum (FBS), penicillin and streptomycin were obtained from Life Technologies, Grand Island, NY. Melanin, 2’,7’-dichlorofluorescein diacetate (DCFH_2_-DA), 4′-6-diamidino-2-phenylindole (DAPI), tyrosinase (EC 1.14.18.1, activity of 6680 units/mg) and 3-(4,5-dimethyl-thiazol-2-yl)-2,5-diphenyl tetrazolium bromide (MTT) were purchased from Sigma-Aldrich (St Louis, CA, USA). Specific pharmacological inhibitors for ERK1/2 (PD98059), p38 MAPK (SB203580), JNK/SAPK (SP600125) and PI3K/AKT (LY294002) were obtained from Calbiochem (La Jolla, CA, USA). Forskolin was obtained from Selleckchem (Houston, TX, USA). Primary and secondary antibodies used in this study were listed in Appendix A. All other chemicals were reagent grade or HPLC grade and supplied by either Merck (Darmstadt, Germany) or Sigma-Aldrich.

### 4.3. Cell Culture and Cell Viability Assay

Murine melanoma (B16-F10), human skin fibroblast (CCD966SK) and human skin keratinocytes (HaCaT) cell lines were obtained from American Type Culture Collection (ATCC, Manassas, VA, USA). Human epidermal melanocytes-adult (HEM-a) was purchased from ScienCell Research Laboratories, Caralsbad, CA, USA. Cells were cultured in Dulbecco’s Modified Eagle medium (DMEM) or Roswell Park Memorial Institute (RPMI) medium or melanocyte medium (MelM), supplemented with 10% fetal bovine serum (FBS), glucose, penicillin and streptomycin. Cells were grown in 10 cm culture dish and incubated in a humidified atmosphere containing 5% CO_2_ at 37 °C. Cell viability was assessed by MTT colorimetric assay as described previously [28].

### 4.4. Determination of Melanin Content and Tyrosinase Activity

Melanin content and tyrosinase activity was determined as described previously [29]. Briefly, B16F10 cells were seeded in 10 cm cell culture dish at a density of 5 × 10^5^ cells/dish. The 50% confluent cells were treated with forskolin (20 μM) in the presence or absence of LEO (25–100 μg/mL) or REO (25–100 μg/mL) or arbutin (100 μM) or kojic acid (20 μM) for 48 h. After treatment, the cells were harvested and washed twice with PBS and the intercellular melanin was solubilized in 1 N NaOH. The melanin content was determined by measuring the absorbance at 475 nm using an enzyme-linked immunosorbent assay (ELISA) micro-plate reader. On the other hand, cells were treated with similar condition for 48 h. The cultured cells were lysed using lysis buffer and clarified by centrifugation at 11,000× *g* for 10 min. 90 µL of each lysate containing an equal amount of protein (100 µg) was placed into a 96-well plate and 10 µL of 15 mM _L_-DOPA was added per well. After incubation at 37 °C for 20 min, the dopachrome formation was measured at 475 nm using an ELISA micro-plate reader. Mushroom tyrosinase activity was determined by ELISA micro-plate reader as described previously [29].

### 4.5. RNA Extraction and Q-PCR Analyses

Total RNA was extracted from cultured B16-F10 cells using the Trizol Reagent (Thermo Fisher Scientific, Waltham, MA, USA). Total RNA concentration was quantified with a NanoVue Plus spectrophotometer (GE Health Care Life Sciences, Chicago, IL, USA). Real-time PCR was performed on a real-time PCR detection system and software (Applied Biosystems, Foster City, CA, USA). First-strand cDNA was generated by SuperScript III reverse transcriptase kit (Invitrogen). Quantification of mRNA expression for genes of interest was performed by qPCR reactions were performed with equal volume of cDNA, forward and reverse primers (10 µM), power SYBR Green Master Mix (Applied Biosystems). The mRNA levels were normalized with GAPDH, while miRNA levels were normalized with U6. The primer sequences of each gene for qPCR were designed by TRIbioteck (Hsinchu, Taiwan) and summarized in Appendix A.

### 4.6. Protein Extraction and Western Blot Analysis

Cells were lysed by either mammalian protein extraction reagent or radio-immunoprecipitation assay (RIPA) buffer (Pierce Biotechnology, Rockford, IL, USA). Protein concentrations were determined using Bio-Rad protein assay reagent (Bio-Rad Laboratories, Hercules, CA, USA). Equal amount of protein samples (60–100 µg) were separated by 8–12% SDS-PAGE and separated proteins were transferred onto polyvinylidene fluoride (PVDF) membrane for overnight. The transferred protein membranes were blocked with 5% non-fat skim milk for 30 min, followed by incubation with specific primary antibodies for overnight and either horseradish peroxidase-conjugated anti-rabbit or anti-mouse or anti-goat antibodies for 2 h. Immunoblots were developed with the enhanced chemiluminescence (ECL) reagents (Advansta Inc., San Jose, CA, USA), images were captured by ChemiDoc XRS+ docking system and the protein bands were quantified by using Imagelab software (Bio-Rad laboratories, Hercules, CA, USA).

### 4.7. Immunofluorescence and Confocal Microscopy

B16-F10 cells at a density of 2 × 10^4^ cells/well were seeded in eight-well glass Nunc Lab-Tek^®^ chamber (ThermoFisher Scientific, Waltham, MA, USA) and treated FRK with or without LEO or REO for indicated time points. After treatment, culture media was removed and the cells were fixed in 2% paraformaldehyde for 15 min, cells were permeabilized with 0.1% Triton X-100 for 10 min, washed and blocked with 10% FBS in PBS and then incubated for 2 h with the anti-phospho-CREB or anti-MITF primary antibodies in 1.5% FBS. The cells were then incubated with the fluorescein isothiocyanate (FITC)-conjugated secondary antibody for another 1 h in 6% bovine serum albumin (BSA). Next, the cells were stained with 1 μg/mL DAPI for 5 min, washed with PBS and visualized using a laser scanning confocal microscope (Leika Microsystems, Buffalo Grove, IL, USA) at a 20 × magnification.

### 4.8. siRNA Transfection

siRNA was transfected with Lipofectamine RNAiMax (Invitrogen) according to the manufacturer’s instructions. For transfection, B16-F10 cells were plated in 6-well plates to give 40–60% confluence at the time of transfection. The next day, the culture medium was replaced with 500 μL of Opti-MEM (GIBCOBRL/Invitrogen) and the cells were transfected using the RNAiMAX transfection reagent. In a separate tube, 100 pM of siRNA was added to 500 μL of an Opti-MEM and RNAiMAX reagent mixture. The resulting siRNA/RNAiMAX mixture was incubated for an additional 25 min at room temperature to allow complex formation. Subsequently, the solution was added to the cells in the 6-well plates, giving a final transfection volume of 1 mL. After 6 h incubation, the transfection medium was replaced with 2 mL of standard growth medium and the cells were cultured at 37 °C. After treatment with LEO or REO for 2 or 48 h, the cells were then subjected to Western blot analysis or melanin quantification.

### 4.9. Immunoprecipitation

B16-F10 cells (1 × 10^6^ cells/dish) were seeded in 10 cm dish and treated with FRK in the presence or absence of LEO or REO or MG132 for 12 h. Cells were collected and centrifuged at 1000× *g* for 5 min and the pellet was resolved in RIPA buffer and then homogenized in an ultrasonicator for 5 s for five times and incubated on ice for 30 min. The lysates were centrifuged at 16,000× *g* for 10 min at 4 °C and the supernatant was recovered. After pre-cleaning of protein A agarose (PAA) with RIPA buffer, protein samples were incubated with PAA for 1 h at 4 °C. Then 3 μg of primary antibody against MITF in antibody diluent (1% BSA, 1% NaN_3_ in PBS) was added to PAA protein complex and incubated for overnight. The PAA, protein and antibody complex was collected by brief centrifugation at 2000× *g* for 5 min and subjected to 3 rounds of washing with 1 mL of RIPA buffer. After adding 50 μL of 3 × sample dye (125 mM Tris-HCL (pH 6.8), 2% SDS, 5% Glycerol, 0.03% bromophenol blue, 1% β-meanptoethanol), samples were resolved by 10% SDS-PAGE and transferred into a PVDF membrane for overnight. The membranes were blocked with 5% skim milk for 30 min and the blots were incubated with antibodies against MITF or ubiquitin for 2 h and then the blots were incubated with horse-radish peroxide (HRP)-conjugated anti-goat IgG or anti-rabbit IgG for 1 h. Immunoblots were developed with the enhanced chemiluminescence (ECL) reagents (Advansta Inc., San Jose, CA, USA), images were captured by ChemiDoc XRS+ docking system and the protein bands were quantified by using Imagelab software (Bio-Rad laboratories, Hercules, CA, USA).

### 4.10. GC–MS Analysis

To determine the chemical composition of leaf and rhizome essential oils of *A. nantoensis*, we carried out GC-MS analyses using an ITQ 900 mass spectrometer coupled with a DB-5MS column as described previously [30]. The temperature program was as follows: 45 °C for 3 min, then increased to 3 °C/min to 180 °C and then increased to 10 °C/min to 280 °C hold for 5 min. The other parameters were injection temperature, 240 °C; ion source temperature, 200 °C; EI, 70 eV; carrier gas, He 1 mL/min; and mass scan range, 40–600 *m/z*. The volatile compounds were identified by Wiley/NBS Registry of mass spectral databases (V. 8.0, Hoboken, NJ, USA), National Institute of Standards and Technology (NIST) Ver. 2.0 GC/MS libraries and the Kovats indices were calculated for all volatile constituents using a homologous series of *n*-alkanes C_9_–C_24_. The major components were identified by co-injection with standards (wherever possible).

### 4.11. Statistical analysis

Data are expressed as mean ± SD. All data were analyzed using the statistical software GraphPad Prism version 6.0 for Windows (GraphPad Software, La Jolla, CA, USA). Statistical analysis was performed using one-way ANOVA followed by Dunnett’s test for multiple comparison. *p* values of less than 0.05 *, 0.01 ** and 0.001 *** were considered statistically significant for the FRK treatment vs. LEO/REO treatment groups. *p* values of less than 0.01 ^#^ was considered statistically significant for the FRK treatment vs. the control group. *p* values of less than 0.05 ^Δ^ was considered statistically significant for the FRK treatment vs. the ERK1/2 inhibitor treatment group.

## 5. Conclusions

From the above data, we conclude that leaf and rhizome essential oils obtained from *A. nantoensis* are able to serve as potential melanogenesis inhibitors. LEO and REO markedly decreased melanin production in FRK-stimulated B16-F10 cells. The effect of LEO/REO on FRK-induced melanin production was attributed to the inhibition of cellular tyrosinase followed by down-regulation of *TYR* and *TRP-1* genes. The effects of LEO/REO on TYR and TRP-1 were might result from suppression of MITF transcriptional activity. Further investigations revealed that LEO and REO regulate MITF through post-translational modification and ubiquitin-mediated proteasomal degradation. LEO and REO-mediated ERK1/2 activation is a key event involved in MITF instability. This study, also unveiled that the secondary compounds α-pinene and _D_-limonene were responsible for their putative anti-melanogenic effects. To the best of our knowledge, this is the first report indicating the _D_-limonene possessed anti-melanogenic properties in vitro. However, further studies are highly warranted to explain the underlying mechanisms of _D_-limonene-mediated anti-melanogenesis. Taken together, these results strongly suggest that LEO and REO could be novel source for the development of skin-whitening agents for cosmetic purposes.

## Figures and Tables

**Figure 1 plants-09-01672-f001:**
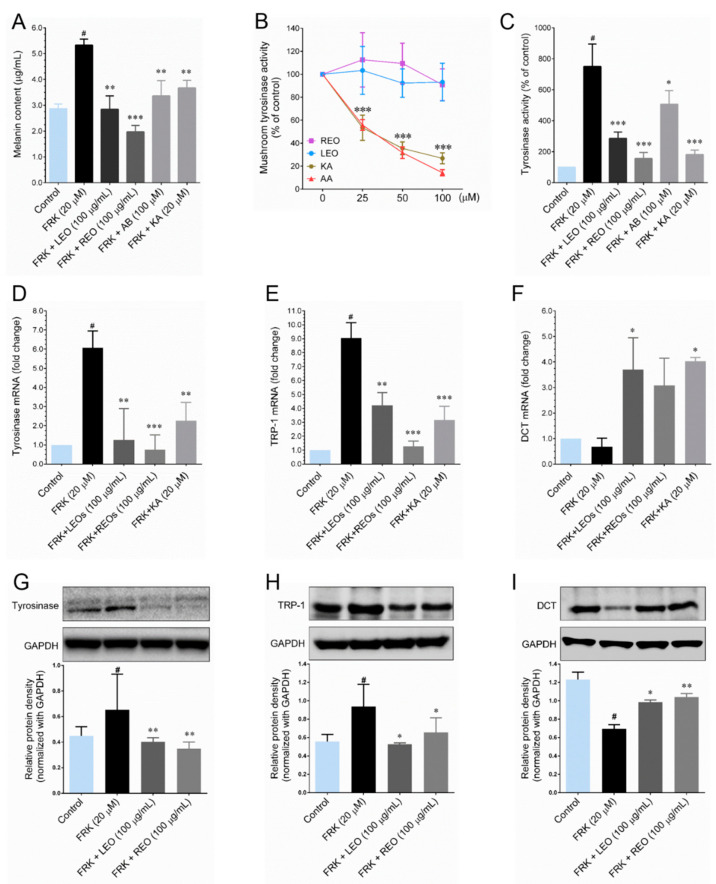
Inhibitory effect of leaf (LEO) and rhizome (REO) essential oils on melanin content and tyrosinase enzyme activity in forskolin (FRK)-induced B16-F10 melanoma cells. (**A**) Cells were treated with indicated concentrations of LEO, REO, arbutin (AB) and kojic acid (KA) and stimulated with FRK for 48 h. Melanin content was assessed by with an absorbance at 405 nm. Cellular melanin content was calculated by comparison with a melanin standard curve. (**B**) The relative activity of mushroom tyrosinase on L-DOPA in the presence of increasing concentrations of LEO, REO, KA and ascorbic acid (AA), compared to the control (100%). (**C**) Effect of cellular tyrosinase activity was determined using whole cell lysates. After treatment with indicated concentration of LEO, REO, AB and KA for 48 h. Cell lysates were used as enzyme source and L-DOPA as substrate. The effects on L-DOPA oxidation velocity was measured at 492 nm. (**D**–**F**) Relative mRNA expression levels of tyrosinase, TRP-1 and dopachrome tautomerase (DCT) were determined by Q-PCR analysis. (**G**–**I**) Protein expression levels of tyrosinase, TRP-1 and DCT were determined by Western blot analysis and the histogram showed relative protein expression, which were normalized with loading control GAPDH. Data represent the mean ± SD of three experiments. Statistical significance was set at # *p* < 0.05 compared to control vs. FRK and * *p* < 0.05, ** *p* < 0.01, *** *p* < 0.001 compared with FRK + sample treatment groups vs. control group.

**Figure 2 plants-09-01672-f002:**
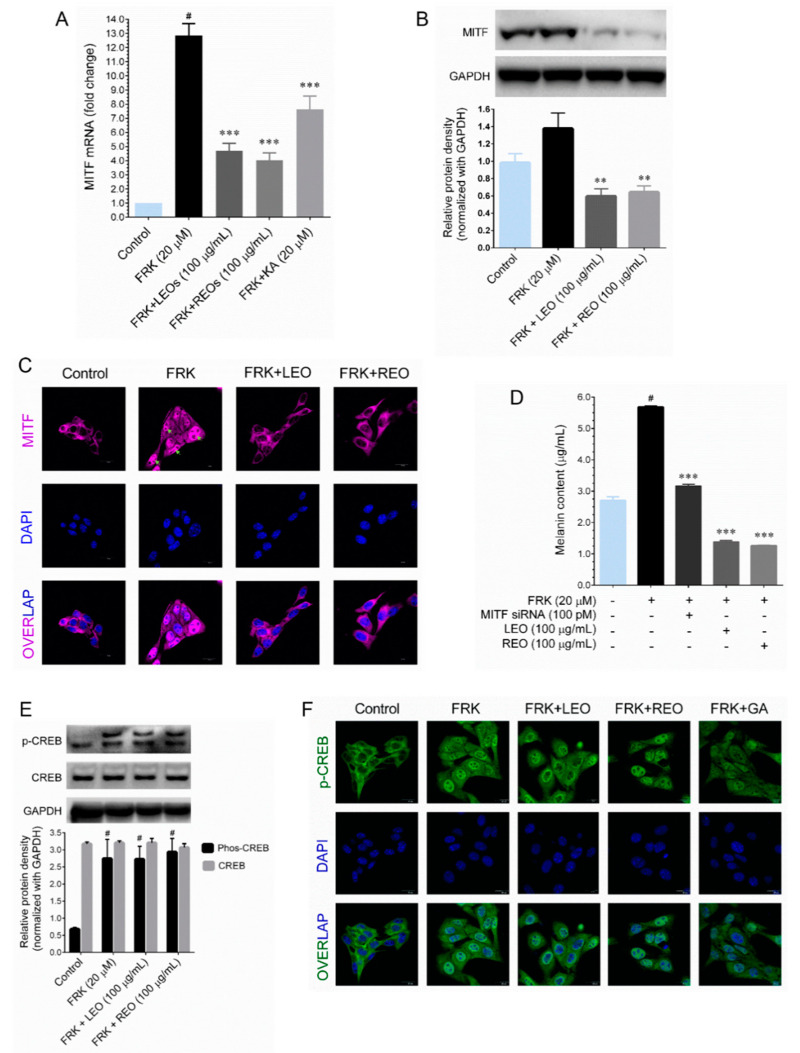
Effect of LEO and REO on Microphthalmia-associated transcription factor (MITF) transcriptional activity. (**A**) The mRNA expression level of MITF was determined by Q-PCR after treatment with 6 h. The protein expression level of MITF was determined by western blot analysis. (**B**) Protein expression level of MITF was determined by Western blot analysis and the histogram showed relative protein expression, which was normalized with loading control GAPDH. (**C**) Cellular localization of MITF was determined by immunofluorescence with fluorescein isothiocyanate (FITC)-conjugated secondary antibody. (**D**) The effect of LEO and REO on melanin content was determined in cells transiently transfected with MITF siRNA or control siRNA. (**E**) The protein expression levels of CREB and Phos-CREB were determined by immunoblotting. (**F**) Cellular localization of Phos-CREB was measured by immunofluorescence after treatment with FRK in the presence or absence of LEO, REO and gallic acid (GA). Data represent the mean ± SD of three experiments. Statistical significance was set at # *p* < 0.05 compared to control vs. FRK and *** *p* < 0.001 compared with FRK + sample treatment groups vs. control group.

**Figure 3 plants-09-01672-f003:**
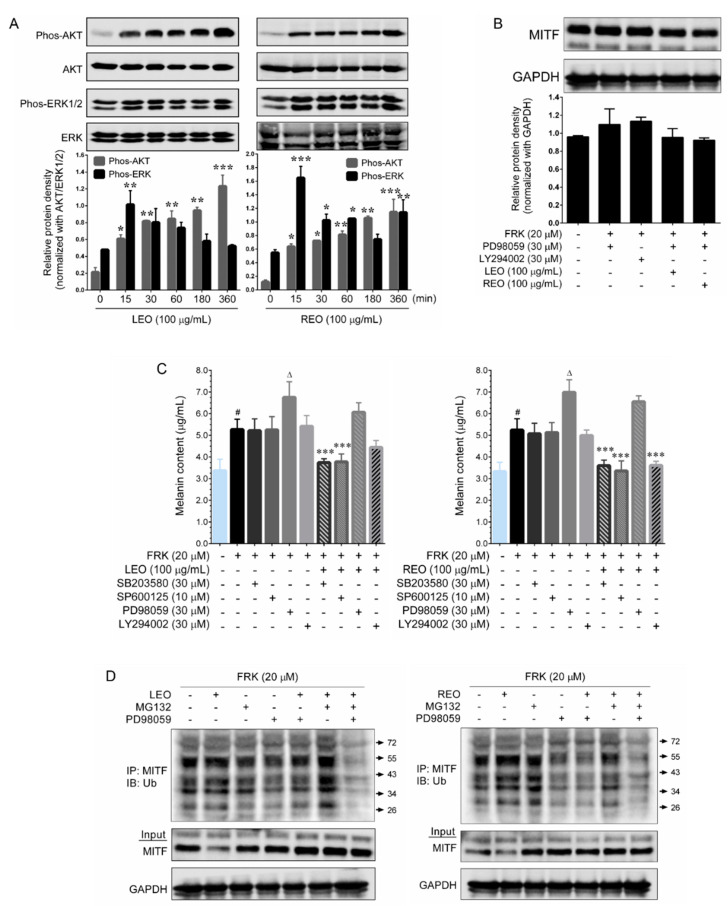
Effect of LEO and REO on ERK1/2 and AKT activation. (**A**) B16-F10 cells were treated with LEO or REO for 15 min to 6 h. The phosphorylation of ERK1/2 and AKT were determined by western blot analysis using specific antibodies. Histogram shows the relative protein expression of phos-ERK1/2 and phos-AKT levels, which are normalized with total levels of corresponding proteins. (**B**) The effect of LEO and REO on MITF protein expression under pharmacological inhibition of ERK1/2 and AKT were determined by western blot analysis. (**C**) Effect of LEO and REO on FRK-induced melanin content was determined after pre-treated with p38 MAPK, ERK1/2, JNK/SAPK and AKT inhibitors for 2 h. (**D**) Ubiquitination of MITF was determined in B16-F10 cells treated with LEO or REO with ERK1/2 and proteasome inhibitors for 6 h. The cells were immunoprecipitated using an anti-MITF antibody and immunoblotted using an anti-ubiquitin antibody. Data represent the mean ± SD of three experiments. Statistical significance was set at # *p* < 0.05 compared to control vs. FRK, Δ *p* < 0.05 compared to FRK vs. ERK1/2 inhibitor (PD98059) and * *p* < 0.05, ** *p* < 0.01, *** *p* < 0.001 compared with FRK + sample treatment groups vs. control group.

**Figure 4 plants-09-01672-f004:**
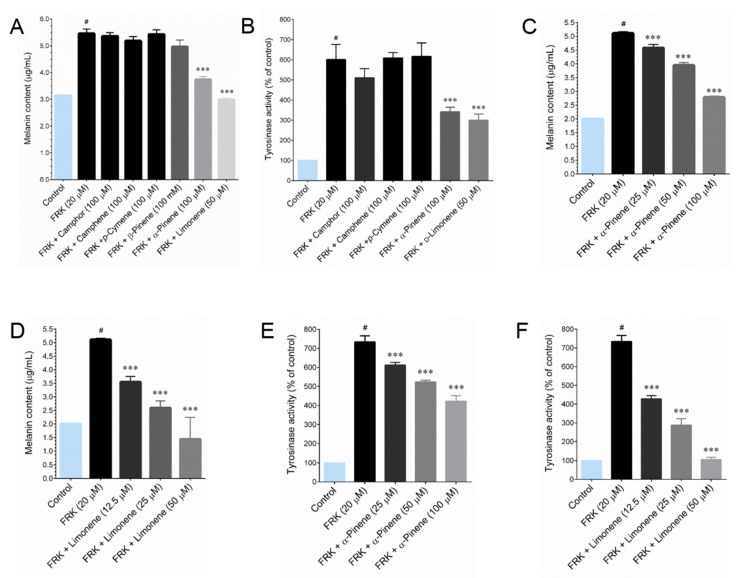
Effect of major compounds of LEO and REO on melanin production and cellular tyrosinase activity in FRK-stimulated cells. (**A**) Cells were treated with indicated concentration of major compounds and stimulated with FRK for 48 h. The melanin content was determined spectrometrically. (**B**,**C**) The effect of α-pinene and D-limonene on FRK-induced melanin production was determined in a dose-dependent manner. (**D**) The effect of cellular tyrosinase activity by major compounds of LEO and REO were determined. (**E**,**F**) Dose-dependent inhibitory effect of α-pinene and D-limonene were quantified. Data represent the mean ± SD of three experiments. Statistical significance was set at # *p* < 0.05 compared to control vs. FRK and *** *p* < 0.001 compared with FRK + sample treatment groups vs. control group.

**Table 1 plants-09-01672-t001:** The major components and their relative contents (%) of leaf and rhizome essential oils of *Alpinia nantonensis*.

Leaf Essential Oil (LEO)						Rhizome Essential Oil (REO)					
No	RT (min)	Compounds	Area (%)	KI	Identification	No	RT (min)	Compounds	Area (%)	KI	Identification
1	10.05	α-Pinene	4.49	933.7	MS, KI, ST	1	10.04	α-Pinene	4.23	933.7	MS, KI, ST
2	10.76	Camphene	11.41	950.3	MS, KI, ST	2	10.76	Camphene	10.45	950	MS, KI, ST
3	12	β-Pinene	8.28	976.8	MS, KI, ST	3	12	β-Pinene	9.13	977	MS, KI, ST
4	13.33	α-Phellandrene	2.06	1003.2	MS, KI	4	13.32	α-Phellandrene	2.24	1003	MS, KI
5	14.2	*p*-Cymene	5.25	1023.5	MS, KI, ST	5	14.19	*p*-Cymene	3.22	1024	MS, KI, ST
6	14.43	_D_-Limonene	2.5	1028.7	MS, KI, ST	6	14.42	_D_-Limonene	2.43	1029	MS, KI, ST
7	14.54	1,8-Cineole	1.77	1031.1	MS, KI, ST	7	14.53	1,8-Cineole	1.77	1031	MS, KI, ST
8	17.86	Linalool	1.17	1097.3	MS, KI, ST	8	20.08	Camphor	29.58	1146	MS, KI, ST
9	20.06	Camphor	21.31	1145.6	MS, KI, ST	9	20.78	Isoborneol	3.03	1161	MS, KI
10	20.79	Isoborneol	3.1	1160.8	MS, KI	10	22.33	α-Terpineol	3.08	1191	MS, KI, ST
11	22.31	α-Terpineol	1.09	1190.6	MS, KI, ST	11	23.37	Fenchyl acetate	1.59	1213	MS, KI
12	32.26	Isocaryophyllene	4	1408.8	MS, KI, ST	12	37.51	Germacrene B	1.42	1540	MS, KI, ST
13	38.76	Caryophyllene oxide	1.37	1571.5	MS, KI, ST	13	40.69	γ-Eudesmol	2.86	1621	MS, KI, ST
14	40.68	γ-Eudesmol	1.21	1621.1	MS, KI, ST	14	41.53	α-Eudesmol	2.88	1645	MS, KI, ST
15	41.54	α-Eudesmol	1.81	1644.5	MS, KI, ST						

KI: Kovats index on a DB-5MS column in reference to *n*-alkanes.

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
