# Peer review of "Essential Oils of *Alpinia nantoensis* Retard Forskolin-Induced Melanogenesis via ERK1/2-Mediated Proteasomal Degradation of MITF"

_plants, 2020, doi:10.3390/plants9121672_

Round 1

Reviewer 1 Report

The present study is well designed and very well prepared. The results are convincing as well.

I clearly recommend this work for publication in Plants.

However, I have 2 minor points to be clarified:

  1. The volume of essential oil extracted from leaf and rhizome dried weight have to be mentioned in the results to have a clear idea of these two resources.
  2. Some herbal essential oils known to be unsuitable for use in skin care products and are likely to cause skin irritation, particularly if you are predisposed to allergic reactions or have extremely sensitive skin conditions. This point is one of the first consideration for the development of a new skin product based on essential oil and a topical or systemic allergic response to this product should always be treated diligently. From my point of view this point should be discussed as well. This does not detract from the quality of this work but it deserves to be mentioned.

Author Response

Reviewer: 1

Reviewer’s opinion: The present study is well designed and very well prepared. The results are convincing as well. I clearly recommend this work for publication in Plants. However, I have 2 minor points to be clarified:

Response:  We would appreciate the reviewer-1, whom provided a positive comments on our work. The reviewer’s comments and our responses as follows.

Comment. 1: The volume of essential oil extracted from leaf and rhizome dried weight have to be mentioned in the results to have a clear idea of these two resources.

Response: Thank you for pointing this out. As per your suggestion, we have included this information in the result section (Sub section 2.1; Page 9, Line 2-3).

Comment. 2: Some herbal essential oils known to be unsuitable for use in skin care products and are likely to cause skin irritation, particularly if you are predisposed to allergic reactions or have extremely sensitive skin conditions. This point is one of the first consideration for the development of a new skin product based on essential oil and a topical or systemic allergic response to this product should always be treated diligently. From my point of view this point should be discussed as well. This does not detract from the quality of this work but it deserves to be mentioned.

Response:  We would appreciate the reviewer for providing a valuable suggestion. We seriously consider your suggestion and also, we would like to mention that we already planned to further extent our understanding on skin lightening effect of LEO and REO on in vivo models. During the investigation, we will definitely examine adverse effects of LEO and REO to the skin tissues.

Reviewer 2 Report

In this article, the author studied that essential oils of Alpinia nantoensis retard forskolin-induced melanogenesis via ERK1/2-mediated proteasomal degradation of MITF. In this study author investigated in vitro anti-melanogenic activity of essential oils of Alpinia nantoensis and their bioactive ingredients. Treatment with leaf and rhizome essential oils A. nantoensis significantly reduced forskolin-induced melanin production followed by down-regulation of tyrosinase (TYR) and tyrosinase related protein-1 (TRP-1) expression at both transcriptional and translational levels. This study provided a strong evidence that LEO and REO could be promising natural sources for the development of novel skin-whitening agents for the cosmetic purposes. Manuscript is very well written and study is very well planned backed by strong data like 1. Studying inhibitory effect of LEO and REO on melanin content and tyrosinase enzyme activity in FRK-induced B16-F10 melanoma cells. 2. Studying the effect of LEO and REO on MITF transcriptional activity. 3. Studying the effect of LEO and REO on ERK1/2 and AKT activation. This study will attract large viewership and can be accepted as it is.

Author Response

Reviewer: 2

Reviewer’s opinion: In this article, the author studied that essential oils of Alpinia nantoensis retard forskolin-induced melanogenesis via ERK1/2-mediated proteasomal degradation of MITF. In this study author investigated in vitro anti-melanogenic activity of essential oils of Alpinia nantoensis and their bioactive ingredients. Treatment with leaf and rhizome essential oils A. nantoensis significantly reduced forskolin-induced melanin production followed by down-regulation of tyrosinase (TYR) and tyrosinase related protein-1 (TRP-1) expression at both transcriptional and translational levels. This study provided a strong evidence that LEO and REO could be promising natural sources for the development of novel skin-whitening agents for the cosmetic purposes. Manuscript is very well written and study is very well planned backed by strong data like 1. Studying inhibitory effect of LEO and REO on melanin content and tyrosinase enzyme activity in FRK-induced B16-F10 melanoma cells. 2. Studying the effect of LEO and REO on MITF transcriptional activity. 3. Studying the effect of LEO and REO on ERK1/2 and AKT activation. This study will attract large viewership and can be accepted as it is.

Response:  We thank the reviewer-2 provided such a positive and constructive comments on our work.  The responses to reviewer’s comments as follows.